# Genetic and Epigenetic Factors Associated with Postpartum Psychosis: A 5-Year Systematic Review

**DOI:** 10.3390/jcm13040964

**Published:** 2024-02-08

**Authors:** Sophia Tsokkou, Dimitrios Kavvadas, Maria-Nefeli Georgaki, Kyriaki Papadopoulou, Theodora Papamitsou, Sofia Karachrysafi

**Affiliations:** 1Research Team “Histologistas”, Interinstitutional Postgraduate Program “Health and Environmental Factors”, Department of Medicine, Faculty of Health Sciences, Aristotle University of Thessaloniki, 54124 Thessaloniki, Greece; stsokkou@auth.gr (S.T.); kavvadas@auth.gr (D.K.); mgeorgaki@cheng.auth.gr (M.-N.G.); kyriakinp@auth.gr (K.P.); thpapami@auth.gr (T.P.); 2Laboratory of Histology-Embryology, Department of Medicine, Faculty of Health Sciences, Aristotle University of Thessaloniki, 54124 Thessaloniki, Greece; 3Environmental Engineering Laboratory, Department of Chemical Engineering, Aristotle University of Thessaloniki, 54124 Thessaloniki, Greece; 4A’ Neurosurgery University Clinic, Aristotle University of Thessaloniki, AHEPA General Hospital of Thessaloniki, 54636 Thessaloniki, Greece

**Keywords:** postpartum psychosis, genetic risk factors, epigenetic risk factors

## Abstract

**Purpose:** Postpartum psychosis (PPP) is a serious mental health illness affecting women post-parturition. Around 1 in 1000 women are affected by postpartum psychosis, and the symptoms usually appear within 2 weeks after birth. Postpartum mental disorders are classified into 3 main categories starting from the least to most severe types, including baby blues, postpartum depression, and postpartum psychosis. **Materials and Methods:** In this systematic review, genetic and epigenetic factors associated with postpartum psychosis are discussed. A PRISMA flow diagram was followed, and the following databases were used as main sources: PubMed, ScienceDirect, and Scopus. Additional information was retrieved from external sources and organizations. The time period for the articles extracted was 5 years. **Results:** Initially, a total of 2379 articled were found. After the stated criteria were applied, 58 articles were identified along with 20 articles from additional sources, which were then narrowed down to a final total of 29 articles. **Conclusions:** It can be concluded that there is an association between PPP and genetic and epigenetic risk factors. However, based on the data retrieved and examined, the association was found to be greater for genetic factors. Additionally, the presence of bipolar disorder and disruption of the circadian cycle played a crucial role in the development of PPP.

## 1. Introduction

Postpartum mental disorders refer to a spectrum of mental health conditions affecting women post-parturition [1]. During the postpartum period, it is estimated that around 85% of women are affected by mood disturbances. The symptoms can either be mild or severe, appearing in the form of depression or anxiety. The postpartum mental illness women experience is divided into 3 main categories, including baby blues, which is also known as postpartum blues; postpartum depression; and postpartum psychosis [2]. Baby blues affect around 50–85% of new mothers [3], and it is a temporary episode that settles when hormone levels return to their original state at approximately 2 weeks [4]. Postpartum depression occurs within 6 weeks post-delivery and affects around 6.5–20% of women, especially adolescent women with premature infants, with symptoms lasting up to 1 year [5]. Moreover, postpartum psychosis is a severe but reversible mental health condition affecting women post-parturition [6]. It is a rarer but a more serious form of postpartum mental depression, affecting only 0.089–2.6% of women every 1000 births. Worldwide, postpartum psychosis occurs in around 12 million to 352.3 million women giving birth [7]. Severe postpartum mental disorders raise concerns regarding both clinical and public health, and urgent medical care must be provided in order to secure the safety of both the mother and infant. However, in some cases, the outcomes are unfortunate. For instance, poor fetal development as well as infant death and maternal suicide are among the most severe outcomes that can occur [8]. One important piece of information that must be thoroughly examined is the mechanism involved in the development of postpartum psychosis along with the degree of association to both genetic and epigenetic factors. This information is necessary to successfully manage the symptoms and protect the health of those directly involved, namely, the mother and the child.

### 1.1. Symptoms of Postpartum Psychosis

The symptoms of PPP usually appear unexpectedly within hours or a few days during the 2 initial weeks after birth. The symptoms include hallucinations, such as auditory, visual, and olfactory sensations, as well as physical sensations that do not exist. Women with PPP tend to be suspicious with fears, thoughts, and beliefs that are not rational and tend to have manic episodes where they feel high and overactive; they tend to excessively and rapidly talk and lack normal inhibitions due to restlessness [6,7]. In contrast, the manic and overactive episodes are accompanied by low mood episodes characterized by signs of depression, withdrawing from social events, a lack of energy, loss of appetite, anxiety, and insomnia. Thus, these women are in a constant fluctuation of manic and hypomanic feelings. Furthermore, they tend to be in a state of confusion [6,7]. In more alarming cases, women tend to have suicidal thoughts and even act poorly with the motive of harming their child [7].

#### 1.1.1. Classifications of Symptoms

Symptoms can be classified into 3 main categories: depressive, manic, and atypical, which is also known as mixed symptoms.

#### 1.1.2. Depressive Symptoms

The depressive category is the most common and most dangerous classification that comprises up to 41% of PPP cases. Evidence reveals that PPP accompanied by depressive symptoms is always a risk factor in women with a tendency to harm themselves and even their child especially during episodes of hallucinations and delusions that “command” them to cause harm [7]. Statistically speaking, the percentage of women causing harm to a child is around 4.5%, which is 5 times greater in comparison to the other 2 classifications. Suicidal rates of up to 5% have been reported for the depressive classification. The symptoms included in the depressive category are hallucinations and delusions; anxiety and panic attacks; loss of appetite; anhedonia, which is loss of enjoyment in things that the individual usually finds pleasant; depression; and thoughts of self-harm and harming their child.

#### 1.1.3. Manic Symptoms

The second most common subtype is the manic category, affecting around 34% of women. Compared with the depressive category, the percentage of women causing self-harm and harming the child is lower, representing 1% of cases. The symptoms of this subcategory include agitation, irritability, aggressive behavior, lack of sleep, constant and rapid talking, and an increased tendency for delusional thoughts [7].

#### 1.1.4. Atypical—Mixed Symptoms

The third category with the lowest percentage of cases compared with the depressive and manic categories is the atypical category, which is also known as the mixed category, with an approximately 25% chance of occurrence. In this category, women tend to be less aware about their environment. They tend to have disorganized behaviors, exhibit unstructured speech, and appear disorientated. They are in a state of confusion and lack the ability to be alerted by situations taking place around them. In certain cases, they are in a state of catatonia [7], which is described as a lack of stimulation to actions occurring in their surroundings; people with catatonia tend to behave strangely and poorly, leading to life-threatening complications [9].

### 1.2. Risk Factors

PPP is considered to have a multifactorial origin, as various risk factors can trigger its development, including a history of PPP in previous pregnancies, a personal or family history of psychosis or bipolar disorder (BD), a history of schizoaffective disorder, and stopping psychiatric medications during pregnancy [10]. Genetic and epigenetic alterations, conditions during childbirth, and sleep deprivation are considered risk factors for the development of PPP.

#### 1.2.1. Genetic Risk Factors

Genetics, epigenetics, neuroactive molecules, psychiatric history, social support, heath history, use of substances, and adverse life events belong under this branch. Regarding genetics, DNA is an essential baseline for the possible development of diseases [11]. Recent research has shown that 49% of individuals diagnosed with PPP have a history of BD [11]. In cases of full blood siblings where one sister developed PPP, which was triggered during childbirth, higher chances of PPP development are noted for the other sister compared with the rate of occurrence in non-related individuals [12].

#### 1.2.2. Epigenetic Risk Factors

Epigenetics is an alteration in gene function that does not affect the DNA sequence. The main type of alteration that takes place is DNA methylation, which is the addition of a methyl group to the DNA responsible for gene transcription and the specificity of the cell [13]. DNA methylation (DNAm) is a chemical modification found on cytosine phosphate-guanine sites (CpG). DNAm is an influencer of gene expression, genomic stability, and the conformation of chromatin [13]. DNAm biomarkers are used for the identification of any possible alterations in individuals with postpartum mental disorders [13].

#### 1.2.3. Childbirth and Sleep Deprivation as a Risk Factor

Childbirth is a strong biological trigger for the development PPP. Sleep deprivation during labor and the period after childbirth is a risk factor for the development of PPP, especially in women who have previously experienced mania as a result of sleep deprivation. Primiparity is the main obstetric risk factor for the development of PPP in contrast with age, which showed no correlation with PPP [3].

### 1.3. Mother–Infant Bonding among Mothers Suffering from PPP

PPP has unfavorable consequences for both the mother and infant. Mothers with PPP tend to have delusional thoughts related to the infant, leading to extreme behaviors. On the one hand, the mother may appear to be overprotective of her child. In contrast, she may be abusive and neglect the care of the child. However, studies have suggested that the degree of severity can exhibit sociocultural differences. An Indian study reviled that 43% of women suffering from PPP had infanticidal thoughts and 36% reported infanticidal behavior compared with a rate of 8% among Dutch women [3].

Although PPP is a more severe form of postpartum mental disorder, PPD has a higher rate of disturbed mother–infant bonding at approximately 57.1% and only affects 1 in 5 women (17.6%) suffering from PPP [14]. Evidence suggests that children born to mothers who suffer from PPD have high risks of developing socioemotional and cognitive problems. Infants participate in daily interactive routines with their mothers, but maternal depression can influence those daily activities in 2 main ways: intrusiveness or withdrawal.

Intrusive mothers usually create a hostile environment that disrupts the infant’s activities. The infant as a result experiences feelings of anger, which then develop into internalized anger and further into a coping mechanism to prevent its mother intrusive behavior [15].

Withdrawn mothers show a lack of interest in their child, which makes it difficult for them to pay attention and understand when their child is unwell. They are disengage with minimal support of the infant’s activities and are mostly unresponsive when it comes to their child’s needs. As a result, children are unable to self-regulate this negative attitude and develop self-regulatory behaviors [15].

#### 1.3.1. Effects on the Time Scale from Embryonic Life to Adulthood

Regarding a time scale starting from the prenatal stages, infants tend to have poor nutritional levels, higher chances of being born prematurely, and have a low birth weight and inadequate prenatal care. The inadequate prenatal care is a result of the lack of maternal care and an inability of expression of maternal affection from mothers suffering from postpartum mental disorders.

#### 1.3.2. Infancy

During the infancy period, infants reveal both cognitive and behavioral issues, such as low cognitive performance; this develops as a result of the mother being restrictive towards her child and not allowing it to participate in the everyday life activities as it normally should [16]. This will lead to negative behavioral outcomes, including anger outburst and passivity as a protective coping mechanism as well as withdrawal and self-regulatory behaviors due to the lack of infant–maternal bonding. The infant does not develop the appropriate arousals and regulated attention required for its proper development leading to future problems in its third phase of development, the toddler stage.

#### 1.3.3. Toddler

The toddler tends to have a passive and non-compliance behaviors; it tends to ignore authority figures, such as parents or teachers at pre-school, and acts as if commands and rules are non-existent [17]. The child finds it difficult to develop a form of autonomy resulting in less social interactions and tends to be isolated from the rest of its peers. Its ability to socialize shows a lack of cognitive performance as well as lack of creativity in comparison with the rest of the children its age.

#### 1.3.4. Childhood

When the child reaches a school-appropriate age, certain new traits develop including attention deficit and hyperactivity disorder (ADHD) and the verbal as well as the full IQ scores are found to be lower compared to the average children of its age with non-depressed mothers. Children of depressed mothers have impaired adaptive functioning as well as affective problems, meaning a set of psychiatric disorders, such as depression or bipolar disorder with different extents of severity. Additionally, anxiety disorders and conduct disorders, with disregard to third parties and an inability to properly behave in a socially acceptable manner, are noted [17].

#### 1.3.5. Adolescents

Lastly, during the adolescent years of their lives, teenagers reveal great academic challenges due to ADHD, learning disorders, and affective disorders, such as depression, anxiety disorders, phobias, and panic disorders involving frequent and unexpected panic attacks [16]. These children even end up experiencing alcohol and substance abuse. Other studies have shown that children born to mothers who suffer from PPD are at higher risk of poor health compared to the rest of the population, including shorter gestation periods, impaired cardiovascular function, increased gastrointestinal infections, reduced weight gain, and lower respiratory tract infections [17,18].

### 1.4. Epidemiology of Postpartum Psychosis

The postpartum period, which is also known as puerperium, is the period post-parturition up to around 6–8 weeks after delivery where the female body goes through certain physiologic changes to bring the body back to its original state before pregnancy was initiated [19]. This stage is considered a high-risk time for the development of psychiatric disorders due to certain triggers taking place during childbirth [7,16]. A UK study revealed that the chances of a psychiatric disorder development was 22 times more likely in the first month post-delivery compared to during and prior to pregnancy. The likelihood of the development of PPP was greater among women who were primiparous, those bearing a child for the first time [7,16]. It is estimated that around 60% of women suffering from PPP have a history of mental disorders that were either managed in the past and have a recurrent effect due to the triggers caused by the pregnancy or were still present during conception as well as during and post-delivery [8,20].

### 1.5. Evaluation and Diagnosis of Postpartum Psychosis

#### 1.5.1. Medical and Social History

When a patient who has recently given birth presents with psychotic symptoms, a thorough medical history as well as a neuropsychiatric evaluation must take place to obtain a correct diagnosis and treatment [21]. Personal as well as family histories of psychiatric illnesses must be taken into account or excluded. Both prenatal and postpartum records must be thoroughly examined to narrow down any possible medical comorbidities, organic causes, and gynecological and obstetric complications, such as pre-eclampsia, eclampsia, previous negative birth outcomes, and current birth complications [21]. It is important to note whether the patient suffered from past psychotic episodes and whether she continued her medication throughout her pregnancy and or resumed it post-delivery. Any history of substance abuse or current stressors, such as financial difficulties, and social as well as support circles should be taken into consideration when it comes to the evaluation of PPP.

#### 1.5.2. Diagnostic and Statistical Manual of Mental Disorders 5th Edition (DSM5)

The Diagnostic and Statistical Manual of Mental Disorders 5th Edition (DSM5) states that severe depressions are diagnosed based on the presence of 5 out of 9 stated symptoms within 2 weeks that the symptoms appear. The possibility that the symptoms are associated with another condition must be overruled in order to have a clearer diagnosis [22].

#### 1.5.3. Lab Examinations

Lab examinations, including a complete blood count (CBC), electrolytes, blood urea nitrogen (BUM), creatinine levels, glucose, vitamin B12, folate, thyroid function tests (TPO and free T4), calcium, urinalysis, urine culture, urine drug screen, liver function tests (LFTs), CT and brain MRI, can be performed to rule out any medical conditions and organic substances that might interfere and appear as psychotic conditions [7,23]. Conditions that might present as psychosis are hyponatremia; hypernatremia; hypoglycemia; and hyperglycemia, including insulin shock and diabetic ketoacidosis. LFTs will help to exclude hepatic encephalopathy, and thyroid function tests for hypothyroidism and hyperthyroidism should be performed. Urine analysis will be used to identify any possible infections, and CT and MRI examinations are used to evaluate the possibility of a stroke, which is a risk factor for women with pregnancy-induced hypertension, preeclampsia, and eclampsia [21,24].

#### 1.5.4. Edinburgh Postnatal Depression Scale (EPDS)

During prenatal care visits, physicians must provide a screening test also known as the Edinburgh Postnatal Depression Scale (EPDS). The 10-question EPDS includes specific questions that help in the determination and diagnosis of PND. The EPDS is used to evaluate and assess the possible existence of PPP. The EPDS consist of 10 multiple-choice questions provided to the mothers to assess the existence and severity of postpartum depression (PPD) and PPP. The scale is used at 6 to 8 weeks post-delivery and should be fully completed by the mother herself. A score greater than 13 is considered positive for depression [25].

## 2. Methodology

This is a systematic review study that followed a PRISMA flow diagram (Figure 1) to study targeted papers on the association of PPP with genetic and epigenetic risk factors. A table was then prepared to simplify the main articles found.

### Materials and Method

A PRISMA flow diagram was followed to narrow down articles based on the following criteria: articles published within a 5-year range between 2019 and 2024 and the type of articles included were either systematic reviews, literature reviews, or meta-analyses. Case reports and trial studies were excluded. The keywords used included postpartum psychosis (PPP), genetic factors, and epigenetic factors. The main databases used were PubMed, Scopus, and ScienceDirect. Additional sources include websites and organizations, such as the NHS, John Hopkins Medicine, American Pregnancy Society, Cleveland Clinic, Frontier, Springer, etc. Any duplicates were removed, and the information found was narrowed based on certain criteria.

## 3. Results

Initially, before the adjustments for advanced settings were made when searching the keywords stated above in the main databases, 2397 records were identified, including 57 records from PubMed, 49 from Scopus, and 2273 from ScienceDirect. After the application of criteria, 2321 records were excluded automatically. The remaining 58 were screened, and 7 duplicates were removed. Thus, a total of 51 studies remained. From those 51 records, 42 were excluded for the following reasons: 3 were animal studies, 1 was only available as an abstract, 23 papers had low relevance to the current study after the whole article was examined, 5 were excluded due low relevance based on the abstracts examined, 6 were excluded due to low relevance based on the title, and 4 studies were excluded based on study type. Thus, 9 studies remained. In addition, 20 studies were found from additional sources, leading to a total of 29 studies when combined with those identified from the main databases.

After examination of the 9 articles obtained from the main databases, a table (Table 1) was generated using Microsoft Word 2023. The table includes the following categories: Study, Genes and Chromosomal Location—Genetic and Epigenetic Factors, Clinical Aspects, and Treatment. The additional 20 articles found from other sources were used as complementary data to support the main articles. After examination and extraction of data from the 9 main articles, the following observations were made.

## 4. Discussion

When the extraction of information from the articles was completed, the following observations were made with regards to the genetic and epigenetic factors, clinical aspects, and methods of treatment.

### 4.1. Genes and Chromosomal Location—Genetic and Epigenetic Factors

First, a number of genes are associated with depressive disorders, such as variations in the 5-HTT serotonin transporter gene and polymorphisms in the STin2.12 allele [8]; SNPs [9]; clock-controlled genes (CCGs) driven by endogenous molecular clocks that regulate rhythmic expression [31]; SLC6A4, COMT, TPH2, FKBP5, MDD1, HTR2A, and MDD2 as well as methylation in genes BDNF, NR3C1, and OXTR [30]; the CCN gene family with elevated CCN2 and CCN3 expression [28]; deletion of the STS gene [27]; methyl transferase-like 13; chromosomal locations 16p13 and 8q24 [8]; mRNA expression of 5α-reductase type I [25], and PRS folate metabolism (MTHFR C677T) [20]. These genes were found to be mainly associated with PPP, BP, manic symptoms, and major depressive episodes. The 5-HTT serotonin transporter gene, linkage of chromosomal locations 16q13 and 8q24 with CCN2 and CCN3, MTHFR C677T variations and methylation in BDNF, NR3C1, and OXTR genes will be discussed further.

#### 4.1.1. Variations in the 5-HTT Serotonin Transporter Gene

Two studies reported that 5-HTT serotonin transporter gene variations are highly associated with PPP, especially in women with bipolar disorder (BD) [8,32]. Specifically, 5 HTTLPR-VNTR is a variant form of the SLC6A4 gene that is located at the human chromosome 17q11.2 [34]. The polymorphism in this location is biallelic with a 44 bp insertion/deletion leading to the formation of two different alleles: the short (S) allele contains deletions and the long (L) allele contains insertions [35]. Studies have revealed that the presence of allele S decreases the transcriptional efficiency of the 5 HTT promoter, leading to lower levels of serotonin transporter binding and uptake. Thus, a greater risk of susceptibility to psychiatric disorders and major depressive episodes, such as PPD and PPP, is observed [35,36].

#### 4.1.2. Linkage of Chromosomal Locations 16q13 and 8q24 with CCN2 and CCN3

Two studies [8,28] discussed the linkage between PPP and the chromosomal locations 16q13 and 8q24. The CCN gene family comprises of 6 members of which CNN2 and CNN3 are included. Elevated CCN2 and CCN3 gene expression in the brain and abnormal maternal behavioral phenotypes have been indicated [28]. More specifically, it has been associated with PPP, BD, sleep deprivation, manic symptoms, and depressive symptoms. Since the CCN3 gene is located at a distance of 138 cm from 16q13 and 8q24, a signal association can be potentially explained regarding the maternal phenotypic behavior [8,28].

#### 4.1.3. MTHFR C677T Variation

Additionally, two studies discussed SNPs [20,32]. In one of the two studies [16], SNPs of the MTHFR gene are discussed. Specifically, the C677T genetic variant is highlighted [20]. The methylenetetrahydrofolate reductase (MTHFR) gene is located in human chromosomal region 1p36.3. The C677T variant is due to the replacement of a cytosine nucleotide base with thymine, leading to the conversion of the valine to alanine at codon 222 [11]. C677T has been found to be associated with PPP, BD, and major depressive episodes [36].

#### 4.1.4. Methylation in the BDNF and NR3C1 Genes

Epigenetics refers to the science that studies external changes in DNA without any alterations in the nucleotide sequences of the DNA, resulting in functional and behavioral changes in the genes and subsequent alterations in protein function. A study identified from the main database [30] states that degree of methylation, which involves the addition of a methyl group, in the BDNF and NR3C1 genes has a positive association with PPP.

#### 4.1.5. DNA Methylation in the OXTR Gene

DNA methylation (DNAm) was found in oxytocin gene receptors (OXTR), reflecting changes in inflammatory cells. Oxytocin (OT) affects the cell by interacting with the oxytocin receptor gene (OXTR), a G-protein coupled receptor that promotes G-protein signal transduction to the nucleus of the cell upon ligand binding. OXTR transcription is regulated by DNA methylation (DNAm) at a group of sites, such as CpGs, which are present within the OXTR exon. Methylation at these sites can result in various conditions, such as autism spectrum disorder, individual variability, unsocial perception, and callous-unemotional traits. In mice for instance deletion of the OXTR led to deficits in maternal behaviors. Thus, OXTR is consider a risk factor for the development of PPD. Studies support interactions between the OXTR genotype at rs53576 and increased risk for comorbid depressive and disruptive behavior disorders, and OXTR DNAm was associated with women with euthymic moods becoming depressed during postpartum periods [33]. Studies have shown evidence of significant interactions among the rs53576 genotype, the degree of methylation at CpG-934 in OXTR, and the presence of prenatal depression in women with PPD. Moreover, women who do not show any signs of depression throughout pregnancy but who carry the rs53576_GG genotype and display high methylation levels in OXTR are three times as likely to develop PPD in comparison to women with lower methylation levels or carrying the rs53576 A allele [37].

### 4.2. Management and Treatment Options

#### 4.2.1. Pharmacological Interventions

When it comes to treatment options, second-generation antipsychotics (SGA) are more favorable compared with first-generation antipsychotics (FGA) due to lower rates of extrapyramidal symptoms and a reduced likelihood of tardive dyskinesia [38].

#### 4.2.2. Lithium

The advisable treatments include pharmacological interventions, such as lithium, which is an FGA, as a treatment option or for prophylaxis. Lithium is used as a standard treatment for BD and psychotic disorders, such as PPP. Lithium reduces excitation of dopamine and glutamate and increases inhibitory GABA neurotransmission. In more severe cases where psychosis is present prior to delivery, lithium is advisable despite being considered harmful to the embryo’s development, especially in extreme cases where the mother is capable of harming herself and the infant and the benefit of taking the medication outweighs the costs [8,33,38,39]. However, it must be noted that lithium comes with high risk factors; thus, certain examinations must be performed before the administration of lithium, such as renal disease screening and thyroid disease analysis, and an electrocardiogram (ECG) should be performed for individuals with coronary risk factors, hypertension, dyslipidemia, and smoking habits [38]. Additionally, if the woman wants to breastfeed her child, lithium might not be the best option. Lithium crosses into breast milk in large quantities and thus may affect the infant [10].

#### 4.2.3. Benzodiazepines and Brexanolone—GABA Inhibitory Neurotransmitter

Another form of medication is antipsychotics, specifically benzodiazepines falling under the FGA category. These drugs are favorable, especially in cases of women suffering from insomnia [21] as they increase inhibitory GABA neurotransmission [40,41]. Additionally, ZULRESSO (Brexanolone) is another medication prescribed for moderate to severe forms of PPD and PPP. Brexanolone is a neurosteroid like allopregnanolone; these drugs are neuroactive GABA inhibitory neurotransmitters that act as receptor modulators [42].

#### 4.2.4. Antidepressants

Antidepressants, such as selective serotonin reuptake inhibitors (SSRIs) as well as norepinephrine reuptake inhibitors, have been suggested as effective treatments for PPD and show favorable results in women suffering from BD [29,30,31].

#### 4.2.5. Neuroleptics

Neuroleptics, such as haloperidol, have been used as treatment options for PPP and BD. However, as haloperidol is an FGA, it is more likely to cause adverse effects; thus, SGA medications are more favorable as a treatment method [31].

#### 4.2.6. SGA Antipsychotics

As stated above, SGA antipsychotics are more favorable than FGA. When it comes to the best choice for use, the patient’s additional health issues and psychiatric symptoms must be taken into account. For patients who cannot take lithium as a treatment option, SGAs are used as an alternative monotherapy. SGAs include olanzapine, quetiapine, risperidone, and clozapine. It has been suggested that both olanzapine and quetiapine are the best options for women who want to breastfeed their children [10,38,43]. Clozapine is considered unsafe while breastfeeding.

#### 4.2.7. Hypnotics

Zopiclone is another treatment option; however, it is important that breast-fed infants are monitored for sedation, hypotonia, and respiratory distress, especially with regular use of large doses of hypnotics [10].

#### 4.2.8. Alternative Forms of Treatment—Non-Pharmacological Agents

Cognitive-behavioral therapy (CBT) involves efforts to change thinking and mindset [44]. Mindfulness-based cognitive therapy (MBCT) prevents relapse of recurring episodes of depression or deep unhappiness [45]. Interpersonal therapy (IPT) is a form of psychotherapy that focuses on relieving symptoms by improving interpersonal functioning [17]. These therapies have been suggested as methods of treatments, but the most common treatment found in 3 of the 9 articles [20,29,30] is electroconvulsive therapy (ECT).

#### 4.2.9. Electroconvulsive Therapy (ECT)

ECT is an alternative form of treatment and is usually performed in patients who do not respond to antipsychotic medication and mood stabilizers [46]. ECT is procedure performed under anesthesia, and it involves the use of small electric currents passing through the brain and causing minor and brief seizures, with the aim of altering the chemical wiring of the brain and thus reversing the symptoms of PPP [47]. It is mostly used to treat depressive episodes, but it is also used for the treatment of schizophrenia. There is a 77% rate of success in schizophrenic patients. Strong evidence was found for improvements in women with PPP after undergoing ECT [39,47]. However, side effects include temporary memory loss of current events, headaches, nausea and brief confusion that does not last longer than a few hours [39,47].

## 5. Conclusions

The aim of this study was to assess the genetic and epigenetic factors associated with PPP. After reviewing the articles, it was found that genetic factors play a more dominant role compared with epigenetic alterations. Genes found to play a significant role include the 5-HTT serotonin transporter gene. Methylation was noted in the following genes: OXTR, the CCN gene family with elevated CCN2 and CCN3 expression, and PRS folate metabolism (MTHFR C677T). Throughout the examination of the articles, high linkage between PPP of BD was observed. Women with a history of BD had a much higher risk of developing PPP, which was triggered during and after childbirth, in comparison with women with no history of BD. The disruption of the circadian rhythm was seen to play an equally important role as clock-controlled genes (CCGs) have been found to affect mood-regulating brain regions, leading to the development of mood disorders [33]. Thus, further investigations are needed to assess the relationship among the 3 factors as well as existing and additional genes that are linked with PPP, BD, and circadian rhythms. ECT might be the last treatment option. However, it is considered as an early option in severe cases to minimize the risk of unwanted outcomes regarding the well-being of the mother and the child [10]. Lastly, lithium is a commonly used approach for both BD and PPP. However, because it is a FGA antipsychotic, evidence suggests that the use of SGA antipsychotics might be a more favorable option as there are lower rates of extrapyramidal symptoms and a reduced likelihood of tardive dyskinesia. In addition, SGA antipsychotics, such as olanzapine and quetiapine, have been suggested as best treatment options while breastfeeding due to lower transmission to the maternal milk compared with lithium [10].

## Figures and Tables

**Figure 1 jcm-13-00964-f001:**
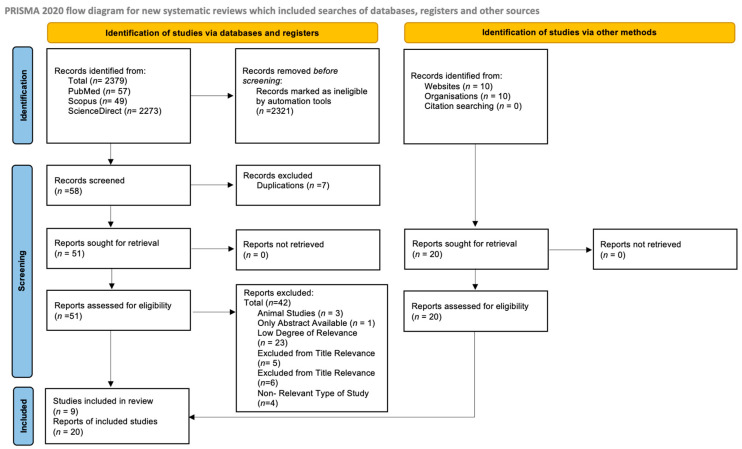
PRISMA Flow Diagram for Postpartum Psychosis and Genetic and Epigenetic Risk Factors. The main databases used include PubMed, ScienceDirect, and Scopus. Studies published during a 5-year period were identified [26].

**Table 1 jcm-13-00964-t001:** Data Collected from the PRISMA Flow Diagram.

Study	Genes and Chromosomal Location-Genetic and Epigenetic Factors	Clinical Aspects	Treatment
Perry A, 2021 [8]	Chromosomal locations 16p13 and 8q24 Methyl transferase-like 13 Variations in the 5-HTTT serotonin transporter gene STin2.12 allele	PPP BD	Lithium
Thippeswamy H, 2021 [27]	Deletion of the STS gene	High risk of developmental disorders and associated traits Increased self-reported irritability Psychological distress Manic symptoms Fatigue and altered sleeping patterns Weight changes—rare cases with paranoid schizophrenia	-
Friedman SH, 2023 [20]	PRS folate metabolism (MTHFR C677T)	PPP Major depression BD	Pharmacological intervention for PPP (Lithium) Antipsychotics (Benzodiazepines) ECT
Davies W, 2019 [28]	CCN gene family (elevated CCN2 and CCN3 levels) Chromosomal locations 16p13 and 8q24	PPP BD Sleep deprivation Manic symptoms Depressive symptoms	-
Sharma V, 2022 [29]	-	BD PPP Major Depressive Episodes Manic Symptoms	Neuroleptics (Haloperidol) Lithium Antipsychotics ECT
Alshaya DS, 2022 [30]	SLC6A4 COMT TPH2 FKBP5 MDD1 HTR2A MDD2 Methylation in BDNF, NR3C1, and OXTR genes	PPP Depression	Antidepressants Selective serotonin reuptake inactivators Serotonin/norepinephrine reuptake inhibitors ECT CBT MBC IPT
Walton NL, 2023 [31]	mRNA expression of 5α-reductase type I	PPP Anxiety disorder PPD BD Schizophrenia	ZULRESSO (Brexanolone) Antidepressants Antipsychotics Mood stabilizers
Silveira PP, 2023 [32]	Serotonin transporter gene polymorphism SNPs	Depression	-
Bhatnagar A, 2023 [33]	CCGs	Depressive-like behavior Mood deficits Manic behavior BD	Lithium Light therapy Melatonin supplements Awake-promoting medications

Abbreviations: Postpartum psychosis (PPP), postpartum depression (PPD), bipolar disorder (BD), clock-controlled genes (CCGS), electroconvulsive therapy (ECT), cognitive-behavioral therapy (CBT), mindfulness-based cognitive therapy (MBCT), interpersonal therapy (IPT), single nucleotide polymorphisms (SNPs).

## Data Availability

The data presented in this study are available on request from the corresponding author Karachrysafi Sofia (email: skarachry@auth.gr).

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
