# Peer review of "Genetic and Epigenetic Factors Associated with Postpartum Psychosis: A 5-Year Systematic Review"

_jcm, 2024, doi:10.3390/jcm13040964_

Round 1
Reviewer 1 Report
Comments and Suggestions for Authors
Thanks for this useful article.
-The introduction should be more complete and it is better to use international diagnostic criteria such as ICD 11 and DSM5-TR to define post partum psychosis.
-The definitions of postpartum psychosis, postpartum depression and post partum blue have changed according to the new criteria, so it is better to use the new definitions preferably from DSM5 in this article and to consult with a psychiatrist.
-The use of antidepressants in PPP is associated with the potential risk of switching to bipolar disorder, and care should be taken in this regard.
Author Response
Response to Reviewer 1 Comments
Dear Sir/Madam,
Thank you for giving us the opportunity to submit a revised version of our manuscript. The time and effort given by you and the other reviewers to provide feedback on our work is greatly appreciated. Based on the suggestions received, necessary modifications have been made to the manuscript to improve its quality and clarity. The changes are highlighted within the manuscript for easy reference.
Thank you for your time and efforts in reviewing our manuscript.
We are committed to making any necessary changes to improve the quality and accuracy of our work.
Sincerely,
Authors
Sophia Tsokkou, Dimitris Kavvadas, Maria Nefeli Georgaki, Kiriaki Papadopoulou, Theodora Papamitsou, Sofia Karachrysafi
Here is a point-by-point response to your comments:
Point 1: The introduction should be more complete, and it is better to use international diagnostic criteria such as ICD 11 and DSM5-TR to define postpartumm psychosis.
Response 1: Thanks for pointing it out.
I am attaching below the exact text that was embedded right into the paper. page [2], Introduction:
“Postpartum mental disorders refers to a spectrum of mental health conditions affecting women post-parturition [1]. During postpartum period it is estimated that around 85% of women are affected by mood disturbances. The symptoms can either be mild or severe, appearing as depression or anxiety. The postpartum psychiatric illness women experience is divided into 3 main categories. The 3 categories include baby blues or also known as postpartum blues, postpartum depression and postpartum psychosis [2]. Baby blues affect around 50-85% of new mothers [3] and it's a temporary episode that settles when hormone levels return to their original state at approximately 2 weeks [4]. Postpartum depression occurs within 6 weeks post-delivery and affects around 6.5-20% of women, especially in adolescent women and premature infants with symptoms lasting up to 1 year [5]..”
Point 2: The definitions of postpartum psychosis, postpartum depression and postpartum blue have changed according to the new criteria, so it is better to use the new definitions preferably from DSM5 in this article and to consult with a psychiatrist.
Response 2: Thanks for pointing it out.
I am attaching below the exact text that was embedded right into the paper.
page [6], Introduction- 1.5 Evaluation and Diagnosis of Postpartum Psychosis
“1.5.1 Medical and Social History
When a patient that has recently given birth is presented with psychotic symptoms a thorough medical history as well as a neuropsychiatric evaluation must take place and thus a correct diagnosis and treatment will be implicated [22]. Personal as well as family history of psychiatric illnesses must be taken into account or overruled. Both prenatal and postpartum records must be thoroughly examined to narrow down any possible medical comorbidities, organic causes and gynaecological and obstetric complications such as pre-eclampsia, eclampsia, previous negative birth outcomes and current birth complications [22]. It’s important to note if the patient was suffering with past psychotic episodes and if she continued her medication throughout her pregnancy and or resumed it post-delivery. Any history of substance abuse or current stressors such as financial difficulties and social as well as support circle should be taken into consideration when it comes to the evaluation of PPP
1.5.2 Diagnostic and Statistical Manual of Mental Disorders 5th Edition (DSM5)
The Diagnostic and Statistical Manual of Mental Disorders 5th Edition (DSM5) states that severe depressions are diagnosed based on the presence of 5 out of 9 stated symptoms within 2 weeks that the symptoms appear. The possibility that the symptoms are associated with another condition must be overruled in order to have a clearer diagnosis [23].
Point 3: The use of antidepressants in PPP is associated with the potential risk of switching to bipolar disorder, and care should be taken in this regard.
Response 3: Thanks for pointing it out.
I am attaching below the exact text that was embedded right into the paper.
page [12], Introduction- 4.2 Management and Treatment Options
Reviewer 2 Report
Comments and Suggestions for Authors
This study aimed to investigate the relationship between the genetic and epigenetic factors regarding the association with postpartum psychosis. The results of the included studies are conflicting. There are several issues that should be considered.
1- There were only a small number of papers investigating each polymorphism or epigenetic marker making it difficult to draw firm conclusions, especially if study results were inconsistent.
2- Table 1 includes the data of 9 studies, what about the data of the other 17 additional studies?
3- Table 1 should include more details about the country of the study and the suggested mechanism of the PPP.
4- What about the quality check and risk of bias in the presented studies?
5- Some of the included studies published in non-peer-reviewed websites making it difficult to draw firm conclusions.
6- Most of the included references are retrieved from websites.
Author Response
Response to Reviewer 2 Comments
Dear Sir/Madam,
Thank you for giving us the opportunity to submit a revised version of our manuscript. The time and effort given by you and the other reviewers to provide feedback on our work is greatly appreciated. Based on the suggestions received, necessary modifications have been made to the manuscript to improve its quality and clarity. The changes are highlighted within the manuscript for easy reference.
Thank you for your time and efforts in reviewing our manuscript.
We are committed to making any necessary changes to improve the quality and accuracy of our work.
Sincerely,
Authors
Sophia Tsokkou, Dimitris Kavvadas, Maria Nefeli Georgaki, Kiriaki Papadopoulou, Theodora Papamitsou, Sofia Karachrysafi
Here is a point-by-point response to your comments:
Point 1: There were only a small number of papers investigating each polymorphism or epigenetic marker making it difficult to draw firm conclusions, especially if study results were inconsistent.
Response 1: Thanks for your comment. The aim of this study was to study the genetic and epigenetic factors in association with PPP. However, after the investigation of the articles it was found that genetic factors play a more dominant role in comparison with epigenetic alterations.
Point 2: Table 1 includes the data of 9 studies, what about the data of the other 17 additional studies?
Response 2: The seventeen studies were used as complementary data to compliment the already existing studies found from the systematic review.
Point 3: Table 1 should include more details about the country of the study and the suggested mechanism of the PPP.
Response 3: The initial draft of the table included some countries however since the articles are not clinical cases, thus specific countries cannot be included. Perhaps in subsequent research, we can conduct a systematic literature search specifically for clinical studies, and meta-analysis of the data, to support the existing paper, and to include information about the country, the individual, the proposed mechanism, etc.
Point 4: What about the quality check and risk of bias in the presented studies?
Response 4: Thank you for your response. Generally, the topic that we choose to analyse did’ not have a lot of information. We recognize that risk of bias assessment helps to establish transparency of evidence synthesis results and findings. A risk of bias assessment is a defining element of systematic reviews and often performed for each included study in the review. Unfortunatelly while we were conducting the research and based on the criteria, we used which included Systematic Reveiws, Literature Reviews and Metanalysis the papers found were only literature. We couldnt find an alternative quality assesment method literature reviews. If you have any recomendations we are open to suggestions that can help us improve our work.Point 5: Some of the included studies published in non-peer-reviewed websites making it difficult to draw firm conclusions.
Point 6: Most of the included references are retrieved from websites.
Response 5 & 6: According to comment 4, there is not appropriate assessment tool for this type of studies included in the final review. The studies taken from websites are from specific organizations (Cleveland Clinic, MGH Center for Women’s Mental Health, American Psychological Association, NCBI bookshelf, Mayo Clinic etc.). Despite this, to further reduce the plate error, References of official articles that have been published in international journals are added to further support our research.
Reviewer 3 Report
Comments and Suggestions for Authors
This is a fascinating subject of debate and research. I am interested both in epigenetics and in psychiatric disorders.
First I'm afraid I have to disagree with your statement below. Psychosis is not always part of a depressive disorder. So you should reconsider the classification below.
"When it comes to postpartum depressive disorders there are 3 main classifications starting from the least to most severe including baby blues, postpartum depression and postpartum psychosis."
You cited number 6 https://my.clevelandclinic.org/health/diseases/24152-postpartum-psychosis.
But if you read carefully this source you will find out that depressive symptoms are part of PPP and psychosis is not a type of depression.
Another issue I found is about PPP diagnostics. From all the tests mentioned below, I can agree with only one test and that is the drug screen. All other tests are not specific for PPP and can easily be influenced by other medical conditions and not by PPP. So please reconsider your statement.
"PPP can be diagnosed by a number of ways, including test analysis of either blood, urine or another form of bodily fluid [6]. These are the easiest forms of tests and include total blood count (CBC), electrolytes, blood urea nitrogen (BUM), creatinine levels and well as glucose, Vitamin B12, folate, thyroid function tests such as TPO and free T4, calcium, urinalysis, urine culture as well as a urine drug screen."
Lithium is an out-of-date therapy that is not recommended anymore because of the side effects during pregnancy. So this should be mentioned as a historical approach but not a currently recommended treatment.
The Conclusion doesn't t seem to support the article. In my opinion, the conclusion is not related to epigenetics, but only to genetic factors.
Comments on the Quality of English Language
Moderate editing of English language required
Author Response
Response to Reviewer 2 Comments
Dear Sir/Madam,
Thank you for giving us the opportunity to submit a revised version of our manuscript. The time and effort given by you and the other reviewers to provide feedback on our work is greatly appreciated. Based on the suggestions received, necessary modifications have been made to the manuscript to improve its quality and clarity. The changes are highlighted within the manuscript for easy reference.
Thank you for your time and efforts in reviewing our manuscript.
We are committed to making any necessary changes to improve the quality and accuracy of our work.
Sincerely,
Authors
Sophia Tsokkou, Dimitris Kavvadas, Maria Nefeli Georgaki, Kiriaki Papadopoulou, Theodora Papamitsou, Sofia Karachrysafi
Here is a point-by-point response to your comments:
Point 1: This is a fascinating subject of debate and research. I am interested both in epigenetics and in psychiatric disorders. First, I'm afraid I have to disagree with your statement below. Psychosis is not always part of a depressive disorder. So you should reconsider the classification below. "When it comes to postpartum depressive disorders there are 3 main classifications starting from the least to most severe including baby blues, postpartum depression and postpartum psychosis."
Point 1: Thanks for pointing it out.
I am attaching below the exact text that was embedded right into the paper. page [1-2], Introduction.
Response 1: “Postpartum mental disorders refers to a spectrum of mental health conditions affecting women post-parturition [1]. During postpartum period it is estimated that around 85% of women are affected by mood disturbances. The symptoms can either be mild or severe, appearing as depression or anxiety. The postpartum psychiatric illness women experience is divided into 3 main categories. The 3 categories include baby blues or also known as postpartum blues, postpartum depression and postpartum psychosis [2]. Baby blues affect around 50-85% of new mothers [3] and it's a temporary episode that settles when hormone levels return to their original state at approximately 2 weeks [4]. Postpartum depression occurs within 6 weeks post-delivery and affects around 6.5-20% of women, especially in adolescent women and premature infants with symptoms lasting up to 1 year [5]]..”
Point 3: You cited number 6 https://my.clevelandclinic.org/health/diseases/24152-postpartum psychosis. But if you read carefully this source you will find out that depressive symptoms are part of PPP and psychosis is not a type of depression.
Response 3: I am attaching below the exact text that was embedded right into the paper. page [1-2], Introduction.
“Moreover, postpartum psychosis is a severe but reversible form of mental health condition affecting women post-parturition [6]. It is a rarer form of postpartum disorder but a more serious form, affecting only 0.089- 2.6% of women every 1,000 births. In a worldwide aspect postpartum psychosis occurs in around 12 million to 352.3 million births [8].]….”
Point 4: Another issue I found is about PPP diagnostics. From all the tests mentioned below, I can agree with only one test and that is the drug screen. All other tests are not specific for PPP and can easily be influenced by other medical conditions and not by PPP. So please reconsider your statement. "PPP can be diagnosed by a number of ways, including test analysis of either blood, urine or another form of bodily fluid [6]. These are the easiest forms of tests and include total blood count (CBC), electrolytes, blood urea nitrogen (BUM), creatinine levels and well as glucose, Vitamin B12, folate, thyroid function tests such as TPO and free T4, calcium, urinalysis, urine culture as well as a urine drug screen."
Response 4: I am attaching below the exact text that was embedded right into the paper. page [6], Introduction - 1.5 Evaluation and Diagnosis of Postpartum Psychosis -1.5.1 Medical and Social History
“When a patient that has recently given birth is presented with psychotic symptoms a thorough medical history as well as a neuropsychiatric evaluation must take place and thus a correct diagnosis and treatment will be implicated [22]. Personal as well as family history of psychiatric illnesses must be taken into account or overruled. Both prenatal and postpartum records must be thoroughly examined to narrow down any possible medical comorbidities, organic causes and gynaecological and obstetric complications such as pre-eclampsia, eclampsia, previous negative birth outcomes and current birth complications [22]. It’s important to note if the patient was suffering with past psychotic episodes and if she continued her medication throughout her pregnancy and or resumed it post-delivery. Any history of substance abuse or current stressors such as financial difficulties and social as well as support circle should be taken into consideration when it comes to the evaluation of PPP..”
Point 5: Lithium is an out-of-date therapy that is not recommended anymore because of the side effects during pregnancy. So this should be mentioned as a historical approach but not a currently recommended treatment.
Response 5: I am attaching below the exact text that was embedded right into the paper. page [12], 4.2.1 Pharmacological Interventions
“When it comes to treatment options second-generation antipsychotics (SGA) are more favorable in contrast with first-generation antipsychotics (FGA) due to lower rates of extrapyramidal symptoms and less likelihood of tardive dyskinesia trigger [40].
4.2.2 Lithium
The advisable treatments include pharmacological interventions such as lithium, a FGA either as a treatment option or a prophylaxis measurement - used as a standard procedure for BD and psychotic disorders such as PPP. Lithium reduces excitation of dopamine and glutamate and increases inhibitory GABA neurotransmission. In more severe cases where psychosis is present prior delivery lithium although being considered harmful for the embryo’s development it is advisable as in extreme cases where the mother is capable of harming herself and the infant the benefit of taking the medication outweighs the costs. [8,33,41, 42]. However, it must be noted that lithium comes with high risk factors and thus certain examinations must be performed before the administration of lithium such as renal disease screening, thyroid disease analysis, an electrocardiogram (ECG) to individuals with coronary risk factors and hypertension, dyslipidaemia and smoking habits must be taken into consideration [40]. Additionally, if the woman wants to breastfeed her child lithium might not be the best option as it crosses into breast milk in large quantities and thus affect the infant [43].”
4.2.5 SGA Antipsychotics
As stated above SGA antipsychotics are more favourable than FGA. When it comes to the best choice for use additional health issues and the psychiatric symptoms the patient is presented with must be taken into account. Patients that cannot take lithium as a treatment option, SGAs are used as an alternative monotherapy option. SGAs include olanzapine, quetiapine, risperidone and clozapine. It has been suggested that both olanzapine and quetiapine are the best options for women that want to breastfeed their children [40, 43, 48]. Clozapine is considered unsafe when it comes to breastfeeding.
4.2.6 Hypnotics
Zopiclone is another treatment option but it’s important that breastfed infants are monitored for sedation, hypotonia and respiratory depression, especially with regular use of large doses of hypnotics [43].
Point 6: The Conclusion doesn't t seem to support the article. In my opinion, the conclusion is not related to epigenetics, but only to genetic factors.
Response 6: The aim of this study was to study the genetic and epigenetic factors in association with PPP. However, after the investigation of the articles it was found that genetic factors play a more dominant role in comparison with epigenetic alterations.
I am attaching below the exact text that was embedded right into the paper. page [13-14], Conclusion
“The aim of this study was to study the genetic and epigenetic factors in association with PPP. However, after the investigation of the articles it was found that genetic factors play a more dominant role in comparison with epigenetic alterations. Genes found to play a significant role include 5-HTT serotonin transporter gene. Methylation in genes OXTR, CCN gene family - elevated CCN2 and CCN3 and PRS Folate Metabolism (MTHFR C677T). Throughout the examination of the articles a high linkage between PPP of BD was observed. Women with history of BD had a much higher risk of developing PPP which was triggered during and after childbirth in comparison with women with no history of BD. The disruption of the Circadian Rhythm was seen to play as equally high role as clock-controlled genes CCG have been found to affect mood-regulating brain regions leading to the development of mood disorders [33]. Thus, further investigation is advised in regard to the relation of the 3 factors and further examination of existing and additional genes that are linked with PPP, BD, Circadian Rhythm. ECT might be the last option treatment but when it comes to severe cases it is considered as an early option to minimize the risk of unwanted outcomes when it comes to the well-being of the mother and the child [43]. Lastly, lithium is a commonly used approach for both BD and PPP, however as it’s a FGA antipsychotic evidence suggest that the use of SGA antipsychotics might be a more favorable option as there are lower rates of extrapyramidal symptoms and less likelihood of tardive dyskinesia triggers as and in addition SGA antipsychotics such as olanzapine and quetiapine have been suggested as best treatment options when it comes to breastfeeding due to lower transmission via the maternal milk in comparison with lithium [43].
.”
Round 2
Reviewer 2 Report
Comments and Suggestions for Authors
The authors address my comments
Reviewer 3 Report
Comments and Suggestions for Authors
I am glad that your paper is now in concordance with the reviewer's suggestions.
In my opinion, you answered all of my expectations.
So now your paper is ready to be published.
Thank you!
Comments on the Quality of English Language
Minor editing of English language required